# INTERROLANG:
# Exploring NLP Models and Datasets
# through Dialogue-based Explanations

**Nils Feldhus**[1]     **Qianli Wang**[2,1]     **Tatiana Anikina**[1,3]
**Sahil Chopra**[3,1]     **Cennet Oguz**[1,3]     **Sebastian Möller**[2,1]

[1]German Research Center for Artificial Intelligence (DFKI)
[2]Technische Universität Berlin, Germany
[3]Saarland Informatics Campus, Saarbrücken, Germany
`{firstname.lastname}@dfki.de`

## Abstract

While recently developed NLP explainability methods let us open the black box in various ways (Madsen et al., 2022), a missing ingredient in this endeavor is an interactive tool offering a conversational interface. Such a dialogue system can help users explore datasets and models with explanations in a contextualized manner, e.g. via clarification or follow-up questions, and through a natural language interface. We adapt the conversational explanation framework TALKTOMODEL (Slack et al., 2023) to the NLP domain, add new NLP-specific operations such as free-text rationalization, and illustrate its generalizability on three NLP tasks (dialogue act classification, question answering, hate speech detection). To recognize user queries for explanations, we evaluate fine-tuned and few-shot prompting models and implement a novel Adapter-based approach. We then conduct two user studies on (1) the perceived correctness and helpfulness of the dialogues, and (2) the simulatability, i.e. how objectively helpful dialogical explanations are for humans in figuring out the model's predicted label when it's not shown. We found rationalization and feature attribution were helpful in explaining the model behavior. Moreover, users could more reliably predict the model outcome based on an explanation dialogue rather than one-off explanations.

*Disclaimer: This paper contains material that is offensive or hateful.*

## 1 Introduction

Framing explanation processes as a dialogue between the human and the model has been motivated in many recent works from the areas of HCI and ML explainability (Miller, 2019; Lakkaraju et al., 2022; Feldhus et al., 2022; Hartmann et al., 2022; Weld and Bansal, 2019; Jacovi et al., 2023). With the growing popularity of large language models (LLMs), the research community has yet to present

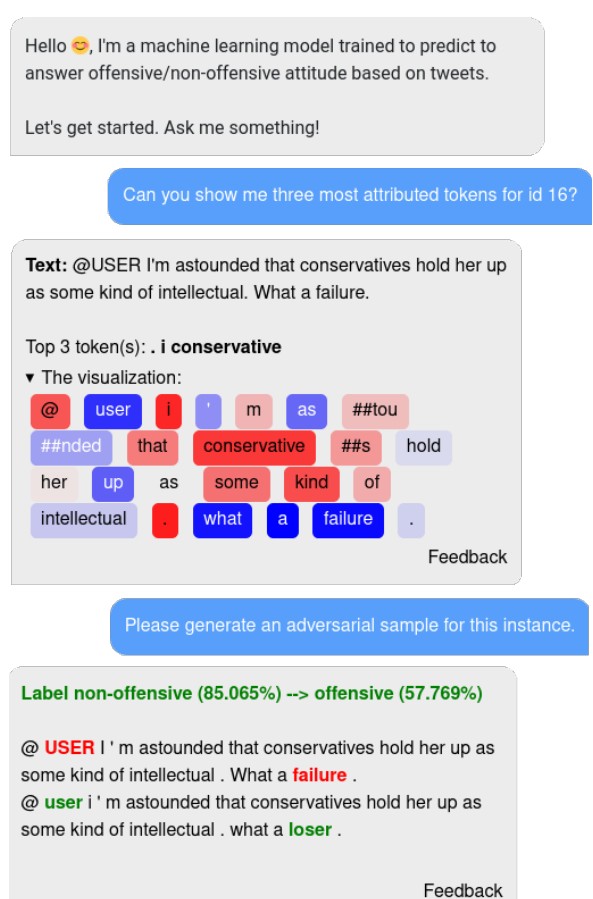

Figure 1: INTERROLANG dialogue with token-level attribution and adversarial example operations on a hate speech detection task (OLID). Users are aware of IDs in the data, since we provide a dataset viewer (not shown).

a dialogue-based interpretability framework in the NLP domain that is both capable of conveying faithful explanations[1] in human-understandable terms and is generalizable to different datasets, use cases and models.

---

[1]While it might be tempting to use ChatGPT, we point out the black-box nature of proprietary software: Most interpretability methods require access to gradients, parameters or training data to make faithful explanations of their behavior. Lastly, it is not possible yet to connect other ML models to it for generating explanations.

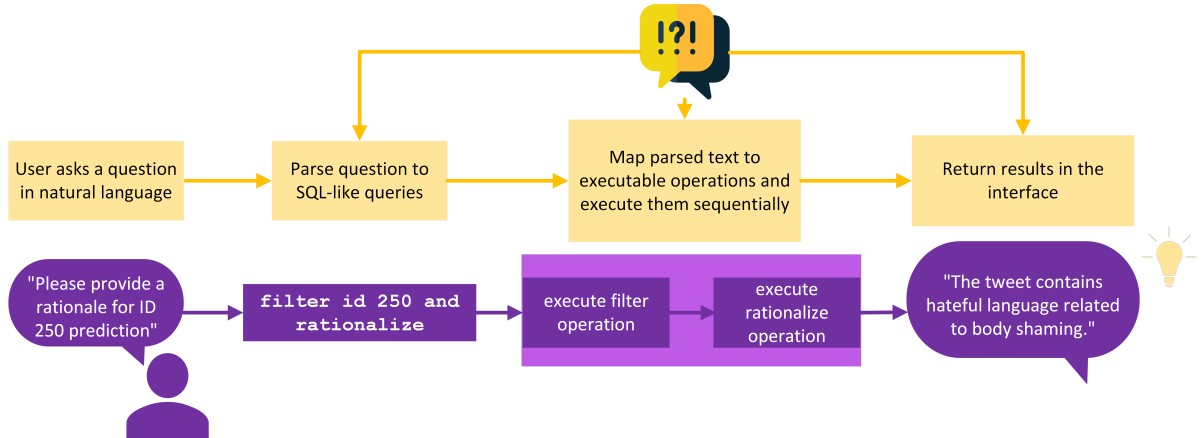

Figure 2: Illustration of how natural language queries from users are parsed into executable operations and their results are inserted in INTERROLANG responses presented through a dialogue interface.

One-off explanations can only tell a part of the overall narrative about why a model "behaves" a certain way. Saliency maps from feature attribution methods can explain the model reasoning in terms of what input features are important for making a prediction (Feldhus et al., 2023), while counterfactuals and adversarial examples show how an input needs to be modified to cause a change in the original prediction (Wu et al., 2021). Semantic similarity and label distributions can shed a light on the data which was used to train the model (Shen et al., 2023), while rationales provide a natural language justification for a predicted label (Wiegreffe et al., 2022). These methods do not allow follow-up questions to clarify ambiguous cases, e.g. a most important token being a punctuation (Figure 1) (cf. Schuff et al. 2022), or build a mental model of the explained models.

In this work, we build a user-centered, dialogue-based explanation and exploration framework, IN-TERROLANG, for interpretability and analyses of NLP models. We investigate how the TALKTO-MODEL (TTM, Slack et al. 2023) framework can be implemented in the NLP domain: Concretely, we define NLP-specific operations based on the aforementioned explanation types. Our system, IN-TERROLANG, allows users to interpret and analyze the behavior of language models interactively. We demonstrate the generalizability of INTERROLANG on three case studies – dialogue act classification, question answering, hate speech detection – for which we evaluate the intent recognition (parsing of natural language queries) capabilities of both fine-tuned (FLAN-T5, BERT with Adapter) and few-shot LLM (GPT-Neo). We find that an efficient Adapter

setup outperforms few-shot LLMs, but that this task of detecting a user's intent is far from being solved. In a subsequent human evaluation (§5.3), we first collect subjective quality assessments on each response about the explanation types regarding four dimensions (correctness, helpfulness, satisfaction, fluency). We find a preference for mistakes summaries, performance metrics and free-text rationales. Secondly, we ask the participants about their impressions of the overall explanation dialogues. All of them were deemed helpful, although some (e.g., counterfactuals) have some potential for improvement. Finally, a second user study on simulatability (human forward prediction) provides first evidence for how various NLP explanation types can be meaningfully combined in dialogical settings. Attribution and rationales resulted in very high simulation accuracies and required the least number of turns on average, revealing a need for a longer conversation than single-turn explanations. We open-source our tool[2] that can be extended to other models and NLP tasks alongside a dataset collected during the user studies including various operations and manual annotations for the user inputs (parsed texts): Free-text rationales and template-based responses for the decisions of NLP models include explanations generated from interpretability methods, such as attributions, counterfactuals, and similar examples.

## 2 Methodology

TALKTOMODEL (Slack et al., 2023) is designed as a system for open-ended natural language dialogues

[2] https://github.com/DFKI-NLP/InterroLang

| | Operation | Description; Question + Explanation example |
|---|---|---|
| | OLID example instance: | *ibelieveblaseyford is liar she is fat ugly libreal snowflake she sold her herself to get some cash !! From dems and Iran ! Why she spoke after JohnKerryIranMeeting ?* |
| **Attribution** | `nlpattribute(instance, granularity)*` | **Desc:** Feature importances on `instance` at (token \| sentence)-level
**Q:** Which tokens are most important?
**E:** *fat*, *ugly* and *liar* are most important for the hate speech label. |
| | `globaltopk(dataset, k, classes)` | **Desc:** Top k most attributed tokens across the entire `dataset`
**Q:** What are the three most important keywords for the hate speech label in the data?
**E:** *dumb*, *fucking*, and *ugly* are the most attributed for the hate speech label. |
| **Perturbation** | `nlpcfe(instance, number)` | **Desc:** Gets number natural language counterfactual explanations for a single `instance`
**Q:** How do you flip the prediction?
**E:** By replacing *liar*, *fat*, *ugly* with neutral nouns and adjectives. |
| | `adversarial(instance)` | **Desc:** Gets number adversarial examples for a single `instance`
**Q:** What is the minimal change needed to cause a wrong prediction?
**E:** *I question the timing of Dr. Ford's statement following the #JohnKerryIranMeeting [...]* |
| | `augment(instance)` | **Desc:** Generate similar `instance`
**Q:** Can you generate one more example like this?
**E:** *I'm skeptical of her integrity and perceive her as a figure manipulated by political agendas.* |
| **Rat.** | `rationalize(instance)` | **Desc:** Explain an `instance` (prediction) in natural language (rationale generation)
**Q:** In natural language, why is this text hateful?
**E:** The text includes multiple instances of insults related to body shaming. |
| **NLU** | `keywords(dataset, number)` | **Desc:** Show most frequent keywords in the dataset
**Q:** What are the most frequent keywords in the dataset?
**E:** *USA*, *president*, *democrats* |
| | `similar(instance, number)*` | **Desc:** Gets number of training data instances most similar to the current one
**Q:** What is an instance in the data very similar to this one?
**E:** *@USER How is she hiding her ugly personality. She is the worst.* |

Table 1: Set of INTERROLANG operations. Descriptions and exemplary question-explanation pairs are added for the hate speech detection use case (OLID). Operations marked with (*) provide support for custom input instances received from users. This applies to single instance prediction as well (Table 8).

for comprehending the behavior of ML models for tabular datasets (including only numeric and categorical features). Our system INTERROLANG retains most of its functionalities: Users can ask questions about many different aspects and slices of the data alongside predictions and explanations. IN-TERROLANG has three main components (depicted in Figure 2): A *dialogue engine* parses user inputs into an SQL-like programming language using either Adapters for intent classification or LLM that treats this task as a seq2seq problem, where user inputs are the source and the parses are the targets. An *execution engine* runs the operations in each parse and generates the natural language response. A *text interface* (Figure 4) lets users engage in open-ended dialogues and offers pre-defined questions that can be edited. This reduces the users' workload to deciding on what to ask, essentially.

## 2.1 Operations

We extend the set of operations in TTM (App. B), e.g. feature attribution and counterfactuals, towards linguistic questions, s.t. they can be used in NLP

settings and on Transformers. In Table 1, we categorize all INTERROLANG operations into Attribution, Perturbation, Rationalization, and Data.

**Attribution** Feature attribution methods can quantify the importance of input tokens (Madsen et al., 2022) by taking the final predictions and intermediate representations of the explained model into account. Next to simple token-level attributions, we can aggregate them on sentence-level or present global top $k$ attributed tokens across the entire dataset (Rönnqvist et al., 2022).

**Perturbation** Perturbation methods come in many forms and have different purposes: We propose to include counterfactual generation, adversarial attacks and data augmentation as the main representatives for this category. While counterfactuals aim to edit an input text to cause a change in the model's prediction (Wu et al., 2021), adversarial attacks are about fooling the model to not guess the correct label (Ebrahimi et al., 2018). Data augmentation replaces spans in the input, keeping the outcome the same (Ross et al., 2022).

**Rationalization** Generating free-text rationales for justifying a model prediction in natural language has been a popular task in NLP (Camburu et al., 2018; Wiegreffe et al., 2022). Such natural language explanations are usually generated by either concatenating the input text with the prediction and then prompting a model to explain the prediction, or by jointly predicting and rationalizing. However, the task has not yet been explored within dialogue-based model interpretability tools.

**Similarity** Inspired by influence functions (Koh and Liang, 2017), this functionality returns a number of instances from the training data that are related to the (local) instance in question. Since influence functions are notoriously expensive to compute, as a proxy, we instead compute the semantic similarity to all other instances in the training data and retrieve the highest ranked instances.

## 2.2 Intent recognition

We follow TTM and write pairs of utterances and SQL-like parses that can be mapped to operations (Table 1) as well as templates that can be filled.

We propose a novel Adapter-based solution (Houlsby et al., 2019; Pfeiffer et al., 2020) for intent recognition and train a model which can classify intents representing the INTERROLANG operations (e.g., `adversarial`, `counterfactual`, etc.). We also train a separate Adapter model for the slot tagging, s.t. for each intent we can label the relevant slots. The slot types that can be recognized by the model include `id`, `number`, `class_names`, `data_type`, `metric`, `include_token` and `sentence_level`. The training details of the Adapter-based approach are listed in Table 9.[3]

The training data for intents are generated from the same prompts that are used for baselines (`GPT-Neo` and `FLAN-T5-base`) with the slot values randomly replaced by the actual values from the datasets (e.g., IDs, class names etc.). Some of the prompts are paraphrased to obtain more diverse training data. Adapter models for intents and slots are fine-tuned on top of the same `bert-base-uncased` model. The performance of

this approach is compared to the prompt-based solution in Table 2.

## 2.3 Dialogue management

We add dialogue management in the form of parsing consecutive operations (Figure 2) and extend it with the ability to handle custom inputs and clarification questions.

TTM, after translating user utterances into a grammar of production rules, composes its results in a template-filling manner while ensuring semantic coherence between multiple operations. They further argue that such a response generation approach prevents hallucinations commonly found in neural networks and conversational models (Dziri et al., 2022). However, it makes the dialogue less natural. That is why we also add a range of pre-defined responses for fallback that are chosen at random when applicable. Moreover, the GPT-based rationales are also the first example of a fully model-generated response. Our system also recognizes when the user just wants to acknowledge the bot's response or intends to finish the conversation and it generates the appropriate responses (see App. H for an example).

When designing dialogue systems, the task of keeping track of the dialogue history is essential to better inform the selection of the next action or response. Thus, we store the previous operations and ids and can resolve deictic expressions like "this sample" or "it" to the ID of the previously mentioned instance. We also check the prediction scores of the intent recognition module to see if there is some problem interpreting the user input, e.g., if several intents get very high scores INTERROLANG asks a clarification question to disambiguate between operations. Also, if we have an intent but some of its non-default slots are missing (not recognized) we can generate a clarification question to resolve it, e.g., "Could you please specify for which instance I should provide a counterfactual?". This gives us more flexibility and makes the dialogue flow more natural.

## 3 NLP Models

We selected three use cases in NLP with BERT-type Transformer models trained on standard datasets, all of which we offer users to explore.

---

[3]Some of the slots are crucial for the intent interpretation and cannot be omitted (e.g., `id` for the `show` operation) while other slots are optional and if not specified by the user the default value is chosen. We also implement additional checks for the case when the user input includes deictic expressions (e.g., "this" in "show me a counterfactual for this sample") in which case the ID of the previous instance is selected.

| Dataset Parsing model | Size | BoolQ | | | OLID | | | DailyDialog | | |
|---|---|---|---|---|---|---|---|---|---|---|
| | | *dev* | *dev-gpt* | *test* | *dev* | *dev-gpt* | *test* | *dev* | *dev-gpt* | *test* |
| Nearest Neighbors | - | 34.69 | 35 | 34.02 | 33.67 | 35 | 30.26 | 36.73 | 37 | 32.51 |
| GPT-Neo | 2.7B | **73** | 70 | 72.54 | 71 | 72 | 67.11 | 70 | 66 | 70.44 |
| FLAN-T5-base | 250M | 71 | 71 | 74.18 | 63 | 66 | 66.67 | 66 | 63 | 75.86 |
| BERT+Adapter | 110M | 72.55 | **76.86** | **79.33** | 72.55 | 76.86 | **84.25** | 72.55 | 77.69 | **83.94** |

Table 2: Exact match parsing accuracy (in %) for the datasets and their three partitions (human-authored *dev* development data, *dev-gpt* data augmented via GPT-3.5, *test* set created from questions asked by participants of the user study). GPT-Neo uses $k = 20$ shots in the prompt.

## 3.1 Dialogue Act classification

DailyDialog (Li et al., 2017) is a multi-turn dialogue dataset that covers different topics related to our daily life (e.g., shopping, discussing vacation trips etc.). All conversations are human-written and there are 13,118 dialogues in total with 8 turns per dialogue on average. We limit the training set to the first 1,000 dialogues, the development set to 100 and the test set to 300 dialogues.

The dialogue act labels annotated in the dataset are as follows: Inform, Question, Directive and Commissive (see Figure 3a for the distribution of labels). Inform is about providing information in the form of statements or questions. Question is used when the speaker wants to know something and actively asks for information. Directives are about requests, instructions, suggestions and acceptance or rejection of offers. Commissives are labeled when the speaker accepts or rejects requests or suggestions (Li et al., 2017). The Transformer model trained on DailyDialog achieves F1 score 68.7% on the test set after 5 epochs of training with 5e-6 learning rate.

## 3.2 Question answering

We choose BoolQ (Clark et al., 2019) as the representative dataset which has been analyzed in the explainability context in many works (DeYoung et al., 2020; Atanasova et al., 2020; Pezeshkpour et al., 2022, i.a.). Each of the 16k examples consists of a question, a paragraph from a Wikipedia article, the title of that article, and a "yes"/"no" answer.

We let its validation set (3.2k instances)[4] be predicted by a fine-tuned DistilBERT (Sanh et al., 2019) model[5] with an accuracy of 72.11%. We choose a smaller model, because it is more easily deployable and more error-prone which increases

the need for explanations.

## 3.3 Hate speech detection

Hate speech detection is a challenging task to determine user entries on social media if offensive. While better models for hate speech detection are continuously being developed, there is little research on the acceptability aspects of hate speech models. There have been a few studies on this task in the explainability literature, mostly using attributions or binary highlights (Mathew et al., 2021; Balkir et al., 2022; Attanasio et al., 2022).

OLID (Zampieri et al., 2019) is one of the common benchmark datasets and includes 14,100 tweets to be identified whether they are offensive. Each row in OLID consists of text and label and the label indicates if the twitter text is "offensive" or "non-offensive". A fine-tuned mbert-olid-en[6] model is used to predict the validation set (2648 instances) and it can achieve an accuracy of 81.42%.

## 4 Interpretability and Analysis Components

For our implementation and experimental setup, we use the following tools and methods to realize the operations in Table 1:

**Attribution** Slack et al. (2023) automatically select "the most faithful feature importance method for users, unless a user specifically requests a certain technique". We constrain feature importance to Integrated Gradients (Sundararajan et al., 2017) saliency scores that we obtain from CAPTUM (Miglani et al., 2023), which allows easy replacement with other saliency methods. The attributions are based on token-level as generated by the underlying model, e.g. BERT in our experiments. We also provide caching functionality to pre-compute and

---

[4]The ground truth labels for the test set are not available.

[5]https://huggingface.co/andi611/distilbert-base-uncased-qa-boolq

[6]https://huggingface.co/sinhala-nlp/mbert-olid-en

store the scores, thus reducing the inference time and mitigating expensive reruns on static inputs.

**Perturbation** For **counterfactual** generation, we use the official Hugging Face implementation of POLYJUICE (Wu et al., 2021)[7]. **Adversarial examples** are generated via OPENATTACK (Zeng et al., 2021)[8], where we choose PWWS (Ren et al., 2019) as the attacker for our models on a single instance. For **data augmentation** we use the NLPAUG library[9] and replace some tokens in the text based on their embedding similarity computed with the *bert-based-cased* model. The percentage of words that are augmented for each text is set to 0.3. We display the replaced words in bold, so that the user can easily distinguish between the original instance and the augmented one.

**Rationalization** As a baseline, we use the parsing model (GPTNeo) in a *zero-shot setup* to produce free-text explanations based on a concatenation of the input, the classification by the explained BERT-type model (Marasovic et al., 2022) and an instruction asking for an explanation. For an improved version, we produce plausible rationales from ChatGPT[10] and then prompt a Dolly-v2-3B[11] for *few-shot* rationales. The rationales are pre-computed for all datasets.

**Natural language understanding** For computing the semantic **similarity**, we embed the data point using Sentence Transformers (Reimers and Gurevych, 2019) and compute the cosine similarity to other points (excluding the instance in question) in the respective dataset. In order to retrieve frequent **keywords** from the whole dataset, we apply the stopwords set defined in NLTK (Bird, 2006) and get a word frequency set. The operation can then return the $n$ most frequent keywords, with $n$ being defined through the user query.

## 5 Evaluation

We conduct our evaluation based on parsing accuracy and two user studies. After introducing the partitions we used to obtain the parsing (intent recognition) results (§5.2), we describe the setup

---

[7]https://huggingface.co/uw-hai/polyjuice
[8]https://github.com/thunlp/OpenAttack
[9]https://github.com/makcedward/nlpaug
[10]https://platform.openai.com/docs/api-reference/chat, March 23 version
[11]https://huggingface.co/databricks/dolly-v2-3b

|  | Operations | Corr. | Help. | Sat. |
|---|---|---|---|---|
| **Metadata** | Show example | 52.94 | 44.44 | 42.19 |
|  | Describe data | 89.66 | 87.27 | 87.72 |
|  | Count data | 56.41 | 44.44 | 45.83 |
|  | True labels | 58.82 | 64.71 | 72.22 |
|  | Model cards | 56.25 | 43.75 | 45.06 |
| **Prediction** | Random prediction | 57.59 | 60.71 | 65.52 |
|  | Single/Dataset prediction | 53.42 | 53.52 | 54.17 |
|  | Likelihood | 62.86 | 67.50 | 63.41 |
|  | Performance | 72.50 | 65.79 | 76.19 |
|  | Mistakes | 81.25 | 68.75 | 77.09 |
| **NLU** | Similar examples | 53.57 | 45.61 | 62.50 |
|  | Keywords | 60.34 | 54.00 | 60.00 |
| **Expl.** | Feature importance | 55.88 | 42.25 | 50.00 |
|  | Global feature importance | 50.00 | 50.00 | 31.32 |
|  | Free-text rationale | 62.07 | 62.50 | 65.45 |
| **Pertb.** | Counterfactual | 40.00 | 27.03 | 21.62 |
|  | Adversarial example | 61.90 | 40.00 | 37.50 |
|  | Augmentation | 62.50 | 52.17 | 60.00 |

Table 3: Task A1 of the user study: Subjective ratings (% positive) on correctness, helpfulness and satisfaction for single turns (responses in isolation), macro-averaged (each user has the same weight, regardless of how many ratings they gave). Custom input operations are averaged with their "regular" counterparts.

of our human evaluation related to user experience and simulatability (§5.3).

### 5.1 Datasets

FLAN-T5-base and Adapter-based models are trained on the *train* set, which contains 505 pairs of user questions and prompts. We automatically extended the set for Adapter by filling in all possible slots with the values from the datasets (Fig. 9). The *train* set is a combination of manual creation by us and subsequent augmentation using ChatGPT. For evaluation, we created three more partitions (*dev*, *dev-gpt*, *test*) to evaluate the parsing accuracy, as presented in Table 2. The *dev* set has been manually created by us which consists of 102 pairs of user questions and parsed texts. To construct the *dev-gpt* set, we leverage ChatGPT to generate semantically similar examples extracted from *dev* set. The *test* set is obtained by collecting questions of participants who participated in the user study (§5.3). Unlike TTM, our NLP datasets don't have a tabular format. Therefore, we had to adjust the parsing approach to be able to handle text inputs relevant to our NLP tasks.

| Datasets | Corr. | Help. | Sat. | Flue. |
|---|---|---|---|---|
| BoolQ | 3.6 | 3.3 | 2.5 | 3.1 |
| OLID | 2.9 | 3.4 | 3.0 | 3.1 |
| DailyDialog | 3.2 | 3.5 | 3.1 | 2.9 |

Table 4: Task A2 of the user study: Subjective ratings (Likert scale 1-5 with 1 being worst/disagree and 5 being best/fully agree) on correctness, helpfulness, satisfaction and fluency for entire dialogues.

## 5.2 Automated evaluation: Intent recognition

To answer the question of how well are user questions mapped onto the correct explanations and responses, for all three use cases, we compare the `GPT-Neo-2.7B` parsing proposed in Slack et al. (2023) with our novel Adapter-based solution (§2.2) and also fine-tune a custom parsing model based on `FLAN-T5-base` (Chung et al., 2022).

## 5.3 Human evaluation

Dialogue evaluation research has raised awareness of measuring flexibility and understanding among many other criteria. There exist automated metrics based on NLP models for assessing the quality of dialogues, but their correlation with human judgments needs to be improved on (Mehri et al., 2022; Siro et al., 2022). While TTM is focused on usability metrics (easiness, confidence, speed, likeliness to use), we target dialogue and explanation quality metrics.

### 5.3.1 Subjective ratings

A more precise way are user questionnaires (Kelly et al., 2009). We propose to focus on two types of questionnaires: Evaluating a user's experience **(1)** with one type of explanation (e.g. attribution), and **(2)** explanations in the context of the dialogue, with one type of downstream task (e.g., QA). An average of the second dimension will also provide a quality estimate for the overall system.

Concretely, we let 10 students with computational linguistics and computer science backgrounds[12] **explore** the tool and test out the available operations and then rate the following by giving a positive or negative review (**Task A**, App. F.1):

1. Correctness (C), helpfulness (H) and satisfaction (S) on the single-turn-level

---

[12]The participants of our user studies were recruited in-house: All of them were already working as research assistants in our institute and are compensated monthly based on national regulations. None of them had any prior experience with the explained models.

| Explanation types | Sim (all) | Sim ($t = 1$) | Help Ratio | #Turns Avg. |
|---|---|---|---|---|
| Local feature importance | 91.43 | 93.10 | **82.86** | 3.85 |
| Sent. feature importance | 90.00 | 94.44 | 60.00 | 3.84 |
| Free-text rationale | **94.74** | **100.00** | 68.42 | **3.70** |
| Counterfactual | 85.00 | 80.00 | 25.00 | 4.14 |
| Adversarial example | 84.00 | 85.71 | 56.00 | 4.00 |
| Similar examples | 88.46 | 87.50 | 61.54 | 4.00 |

Table 5: Task B of the user study: Simulatability. Simulation accuracy (in %), simulation accuracy for explanations deemed helpful (in %), helpfulness ratio (in %), average number of turns needed to make a decision.

2. CHS and Fluency (F) on the dataset-level (when finishing the dialogue)

### 5.3.2 Simulatability

We also conduct a simulatability evaluation (**Task B**, App. F.2), i.e. based on seeing an explanation and the original model input for a previously unseen instance. If a participant can correctly guess what the model predicted for that particular instance (which can also be a wrong classification) (Kim et al., 2016), the explanation they saw would be deemed helpful. We can then express an objective quality estimate of each type of explanation in terms of simulation accuracy, both in isolation and in combination with other explanations.

Each participant (four authors of this paper + two students from Task A) received nine randomly chosen IDs (three from each dataset). The list of operations (Table 5) is randomized for each ID, serving as the itinerary. After each response, the participant can decide to either perform the simulation (take the guess) or continue with the next in the list. After deciding on a simulated label, they are tasked to assign one helpfulness rating to each operation: 1 = helpful; -1 = not helpful; 0 = unused. Let $R$ be the set of all ratings $r_i \neq 0$ and $\mathbf{1}_t(x)$ our indicator function. We then calculate our Helpfulness Ratio as follows:

$$\text{Helpfulness Ratio} = \sum_{r \in R} \frac{\mathbf{1}_1(r)}{|R|}.$$

Let $\hat{y}_i$ be the model prediction at index $i$ and $\tilde{y}_i$ the user's guess on the model prediction, then the simulation accuracy is

$$\text{Sim(all)} = \sum_{i=1}^{|R|} \frac{\mathbf{1}_{\hat{y}_i}(\tilde{y})}{|R|}.$$

Filtering for all cases where the operation was deemed helpful:

$$\text{Sim}(t = 1) = \sum_{i=1}^{|R|} \frac{\mathbf{1}_{\hat{y}_i}(\tilde{y}_i) \cdot \mathbf{1}_t(r_i)}{\mathbf{1}_t(r_i)}.$$

## 6 Results and discussion

**Parsing accuracy** Table 2 shows that our Adapter-based approach (slot tagging and intent recognition) is able to outperform both the GPT-Neo baseline and the fine-tuned FLAN-T5 models, using much fewer parameters and trained on the automatically augmented prompts with replaced slot values.

**Human preferences** Table 3 reveals that most operations were positively received, but there are large differences between the subjective ratings of operations across all three aspects (CHS). We find that data description, performance and mistakes operations consistently perform highly, indicating that they're essential to model understanding. Among the repertoire of explanation operations, free-text rationale scores highest on average, followed by augmentation and adversarial examples, while counterfactuals are at the bottom of the list. The POLYJUICE GPT was often not able to come up with a perturbation (flipping the label) at all and we see the largest potential of improvement in the choice for a counterfactual generator. The dialogue evaluation in Table 4 also solidifies the overall positive impressions. While BoolQ scored highest on Correctness, DailyDialog was the most favored in Helpfulness and Satisfaction. Fluency showed no differences, mostly because the generated texts are task-agnostic. Satisfaction was lowest across the three use cases. Although the operations were found to be helpful and correct, the satisfaction still leaves some room for improvements, likely due to high affordances (too much information at once) or low comprehensiveness. A more fine-grained evaluation (Siro et al., 2022) might reveal whether this can be attributed to presentation mode, explanation quality or erroneous parses.

**Simulatability** Based on Table 5, we can observe that the results align with the conclusions drawn from Table 3. Specifically, free-text rationales provide the most assistance to users, while feature importance was a more useful operation for multi-turn simulation, compared to single-turn helpfulness ratings. On the other hand, counterfactual and adversarial examples are found to be least helpful, supporting the findings of Task A. Thus, their results may not consistently satisfy users' expectations. We detected very few cases where one operation was sufficient. Combinations of explanations are essential: While attribution and rationales are needed to let users form their hypotheses about the model's behavior, counterfactuals and adversarial examples can be sanity checks that support or counter them (Hohman et al., 2019). With $\text{Sim}(t=1)$, we detected that in some cases the explanations induced false trust and led the users to predict a different model output.

### 6.1 Dataset with our results

We compile a dataset from (1) our templates, (2) the automatically generated explanations, and (3) human feedback on the rationales presented through the interface. The research community can use these to perform further analyses and train more robust and human-aligned models. We collected 1449 dialogue turns from feedback files (Task A) and 188 turns from the simulatability study (Task B). We provide a breakdown in App. G.

## 7 Related Work

**Dialogue systems for interpretability in ML** Table 6 shows the range of existing natural language interfaces and conversational agents for explanations. Most notably, CONVXAI (Shen et al., 2023) very recently presented the first dialogue-based interpretability tool in the NLP domain. Their focus, however, is on the single task of LLMs as writing assistants. They also don't offer dataset exploration methods, their system is constrained to a single dataset (CODA-19) and they have not considered free-text rationalization, which we find is one of the most preferred types of operations. Dalvi Mishra et al. (2022) proposed an interactive system to provide faithful explanations using previous interactions as a feedback. Despite being interactive, it does not provide feasibility of generating rationales on multiple queries subsequently. Bertrand et al. (2023) wrote a survey on prior studies on "dialogic XAI", while Fig. 6 of Jacovi et al. (2023) highlights that interactive interrogation is needed to construct complete explanation narratives: Feature attribution and counterfactuals complement each other, s.t. the users can build a generalizable mental model.

**Visual interfaces for interpretability in NLP** LIT (Tenney et al., 2020), AZIMUTH (Gauthier-Melançon et al., 2022), IFAN (Mosca et al., 2023) and WEBSHAP (Wang and Chau, 2023) offer a broad range of explanations and interactive analyses on both local and global levels. ROBUSTNESS GYM (Goel et al., 2021), SEAL (Rajani et al., 2022), EVALUATE (von Werra et al., 2022), INTERACTIVE

MODEL CARDS (Crisan et al., 2022) and DATALAB (Xiao et al., 2022) offer model evaluation, dataset analysis and accompanying visualization tools in practice. There are overlaps with INTERROLANG in the methods they integrate, but none of them offer a conversational interface like ours.

**User studies on NLP interpretability** Most influential to our study design are simulatability evaluations (Hase and Bansal, 2020; Nguyen, 2018; González et al., 2021; Arora et al., 2022; Das et al., 2022; Feldhus et al., 2023). In terms of preference ratings, Strout et al. (2019) evaluated how extractive rationales (discretized attributions) from different models are rated by human annotators. Helpfulness and satisfaction ratings were used in Schuff et al. (2020) and Ray et al. (2019).

## 8 Conclusion

We introduce our system, INTERROLANG, which is a user-centered dialogue-based system for exploring the NLP datasets and model behavior. This system enables users to engage in multi-turn dialogues. Based on the findings from our conducted user study, we have determined that one-off explanations alone are usually not sufficient or beneficial. In many cases, users may require multiple explanations to obtain accurate predictions and gain a better understanding of the system's output.

Future work includes making the bot more proactive, so that it can suggest new operations related to the user queries. We also want to investigate the feasibility of using a singular LLM for all tasks (parsing, prediction, explanation generation[13], response generation) over the modular setup that we currently employ; Redesigning operations as API endpoints and training LLMs to call them (Lu et al., 2023; Schick et al., 2023), s.t. they can autonomously take care of the entire dialogue management at once. Lastly, refining language models (increasing faithfulness or robustness, aligning with user expectations) through dialogues has gained traction (Lee et al., 2023; Madaan et al., 2023). While we are already collecting valuable data, our framework misses an automated feedback loop to iteratively improve the models.

---

[13]Operations have to be adapted in some cases, e.g., generating matrices for feature attribution (Sarti et al., 2023) and counterfactuals without an external library (Chen et al., 2023).

## Limitations

INTERROLANG does not exhaust all interpretability methods, because understanding and integrating them requires a lot of resources. We see feature interactions, measurements of biases and component analysis as the most promising future work.

INTERROLANG does not allow direct model comparison. The models are constrained to their datasets and the use cases are intended to be explored separately.

Users can enter custom inputs to get predicted and explained, but they can not modify the dataset on-the-fly, e.g., adding generated adversarial examples or augmentations directly to the current dataset and saving the updated version.

We do not offer a solution to mitigate biases or potential harmful effects of language models, but INTERROLANG with its range of explanations is intended to point users into directions where the training data or model behavior is counter-intuitive.

We use `ChatGPT` only for (1) producing high-quality rationales to use in demonstrations (§4) and (2) augmenting our intent recognition training data containing utterance-parse pairs (§2.2). We argue that these are legitimate use cases of `ChatGPT`. For almost every other part of INTERROLANG, `ChatGPT` is not applicable, though (see Footnote 1). INTERROLANG is a modular system and one of our goals is to have all modules be sourced from readily available tools. `ChatGPT` can easily be swapped with a sufficiently strong rationalizer and data augmenter, as soon as they become available open source. At the time of implementing INTERROLANG, however, we found that there is a large qualitative gap between `ChatGPT` and open-source LLMs (`Dolly`, `GPT-Neo`) and that's why we opted to include it in these two parts of our framework.

## Ethics Statement

We incorporate `OLID` as one of our datasets, which may contain hateful or offensive words. However, it is important to note that we do not generate any new content that is hateful or offensive. Our usage of the `OLID` dataset is solely for the purpose of assessing the integration of the hate speech detection task to our system and generating plausible and useful explanations.

## Acknowledgments

We are indebted to Gokul Srinivasagan, Maximilian Dustin Nasert, Ammer Ayach, Christopher

Ebert, Urs Alexander Peter, David Meier, João Lucas Mendes de Lemos Lins, Tim Patzelt, Elif Kara and Natalia Skachkova for their invaluable work as annotators. We thank Leonhard Hennig, Malte Ostendorff, João Lucas Mendes de Lemos Lins and Maximilian Dustin Nasert for their review of earlier drafts and the reviewers of EMNLP 2023 for their helpful and rigorous feedback. This work has been supported by the German Federal Ministry of Education and Research as part of the projects XAINES (01IW20005) and CORA4NLP (01IW20010) and the European Union as part of the AviaTor project (SEP-210730802).

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

## A    Explanatory dialogue systems

Table 6 and Table 7 show the range of existing natural language interfaces and conversational agents for explanations.

## B    TALKTOMODEL operations

Most TTM operations belonging to their ML, Conversation and Description categories can be trivially adapted. Here, we document the changes:

Due to Transformers being explained instead of the much smaller scikit-learn models, we applied small changes such as pre-computing predictions (similar to the tricks we used for attributions and rationales).

**Metadata**    For metadata, we provide an operation following the basic idea of model cards (Mitchell et al., 2019) which supplies information related to model details, intended use of the model, etc., and, analogously, datasheets (Gebru et al., 2021) for training/test data documentation. User questions can target specific aspects of this structured information and the system replies in natural language and/or tabular formats.

Table 8 shows the rest of the INTERROLANG operations not depicted by Table 1.

## C    Label distributions of NLP use cases

Figure 3 shows the label distributions of DailyDialog, OLID and BoolQ.

## D    Adapter training details

Table 9 shows the hyperparameters and training time for the Adapter models for dialogue act classification and slot tagging.

| | | Task data | | Model |
|---|---|---|---|---|
| **Implementations** | Num | CV | NLP | |
| DR_ANT (Kuźba and Biecek, 2020) | ■ | | | RF |
| ERIC (Werner, 2020) | ■ | | | DT |
| Torri (2021) | ■ | | | RF |
| TALKTOMODEL (Slack et al., 2023) | ■ | | | RF |
| XAGENT (Nguyen et al., 2023) | ■ | ■ | | RF, CNN |
| CONVXAI (Malandri et al., 2022) | ■ | | | DT, RF |
| CONVXAI (Shen et al., 2023) | | | CODA-19 | Tf |
| **INTERROLANG (ours)** | | | BoolQ DailyDialog OLID | Tf |

Table 6: Explananda (Task and model) comparison of existing implementations of natural language interfaces and conversational agents for XAI. We can see that applications to NLP tasks have started to surface only recently. Task data Num = Numeric/Tabular. CV = Computer vision. Explained model AOG = And-Or graph. DT = Decision Tree. RF = Random Forest. CNN = Convolutional neural network. Tf = Transformer.

| | Explanation types | | | | | Intent recognition / Parsing of user questions | | | | Resp | DST | Evaluation | |
|---|---|---|---|---|---|---|---|---|---|---|---|---|---|
| **Implementations** | FA | CF | Mt | Sim | RG | Comm | Embeds | Fine-Tuned | Few-Shot | | | Auto | Hum |
| Kuźba and Biecek (2020) | ■ | ■ | | | | DiF | | | | DiF | DiF | | |
| Werner (2020) | ■ | ■ | | | | | fastText | | | Rule | | | |
| Torri (2021) | ■ | ■ | | | | | | GPT-2 | | Rule | | | Like |
| Slack et al. (2023) | ■ | ■ | ■ | | | | MPNet | T5 | GPT-Neo/-J | Rule | Rule | ExM | Like |
| Nguyen et al. (2023) | ■ | ■ | ■ | | | | SimCSE | | | Rule | | ExM, F1 | |
| Malandri et al. (2022) | ■ | ■ | ■ | | | RASA | | | | Rule | Rule | | Like |
| Shen et al. (2023) | ■ | ■ | ■ | ■ | | | SciBERT | | | Rule | Rule | | |
| **INTERROLANG (ours)** | ■ | ■ | ■ | ■ | ■ | | MPNet | BERT+Adap, FLAN-T5 | GPT-Neo | Rule | Rule, Adap | ExM | Like |

Table 7: Explanans (XAI modules) comparison of existing implementations of natural language interfaces and conversational agents for XAI. Explanation types FA = Feature Attribution. CF = Counterfactual Generation.

Mt = Meta information about the model. Sim = Similar examples. RG = Rationale generation. Intent recognition Comm = Commercial product (RASA = RASA NLU; DiF = Google DialogFlow). Embeds = Nearest neighbor based on sentence embedding. Response generation / Dialogue state tracking Rule = Rule- and template-based response. Evaluation : **Auto**mated: ExM = Exact match accuracy. **Human**: Like = Likert-scale rating.

| | | |
|---|---|---|
| **Filters** | `filter(id)` | Access single instance by its ID |
| | `includes(token)` | Filter instances by token occurrence |
| **Prediction** | `predict(instance)*` | Get the prediction of the given instance |
| | `predict(dataset)` | Get the prediction distribution across the dataset |
| | `likelihood(instance)` | Obtain the given instance's probability for each class |
| | `mistakes(dataset)` | Count number of wrongly predicted instances |
| | `score(dataset, metric)` | Determine the relation between predictions and labels |
| **Data** | `show(list)` | Showcase a list of instance |
| | `countdata(list)` | Count number of instances within the given list |
| | `label(dataset)` | Describe the label distribution across the dataset |
| **Meta** | `data(dataset)` | Information related to training/test data |
| | `model()` | Metadata of the model |
| **About** | `function()` | Inform the functionality of the system |
| | `self()` | Self-introduction |
| **Logic** | `and(op1, op2)` | Concatenation of multiple operations |
| | `or(op1, op2)` | Selection of multiple filters |

Table 8: TTM operations used in INTERROLANG. *Prediction operation provides support for custom input instances received from users.

# E  Interface

We extend the TTM interface (Slack et al., 2023) in the following ways:

- **Custom inputs**: Compared to TTM, which only allows user to use instances from three predefined datasets, we provide a selection box that allows individual inputs from the user to be considered.
- **Text search**: A search engine that allows the user to filter the dataset according to strings. If a query is present, subsequent operations will consider the subset where this filter is applicable.
- **Dataset viewer**: This shows the first ten instances of the dataset (their IDs and the contents of the text fields) at the start, but in order to make the navigation through the data easier for the user, it will update according to both string filters and

| Parameters | Dialogue Act Classification | Slot Tagging |
|---|---|---|
| Base Model | *bert-base-uncased* | *bert-base-uncased* |
| Learning Rate | 1e-4 | 1e-3 |
| Number of Epochs | 10 | 8 |
| Batch Size | 32 | 32 |
| Optimizer | AdamW | AdamW |
| Number of Labels | 23 | 15 |
| Avg. Training Time | 53 min | 32 min |
| Avg. Model Size | 3.6MB | 3.6MB |
| Training Set | 39,635 | 3,810 |
| Development Set | 11,010 | 635 |

Table 9: Training parameters for the Adapter-based parsing models. The best performing model was selected based on the loss on the development set. All samples are based on the original prompts automatically augmented through the slot value replacements.

operations like label filters.

## F    Annotation instructions

### F.1    Task A

Figure 5 and Figure 6 show the instructions of the user study on subjective ratings (Task A) as described in §5.3.1. Figure 7 shows a screenshot of the Google Forms in Task A2.

### F.2    Task B

Figure 8 shows the instructions of the user study on simulatability described in §5.3.2.

## G    INTERROLANG Dataset statistics

Across all three datasets we have 659 unique user questions that don't overlap with the INTERROLANG sample prompts (81.16%) and 153 questions that do overlap. The high number indicates that our prompts approximate the actual user questions rather well. On the other hand, some of the user questions were taken directly from the prompt examples.

In particular, OLID has 180 (61.2%) unique user questions with 114 overlaps; DailyDialog has 208 (69.3%) unique user questions and 92 overlaps; BoolQ has 192 (88.1%) unique user questions, 26 overlaps. Across all three datasets this results in 478 unique questions (58.9%) and 334 overlapping ones.

(a) Dialogue Act Distribution

(b) OLID Distribution.

(c) BoolQ Distribution

Figure 3: Label distribution of all three datasets.

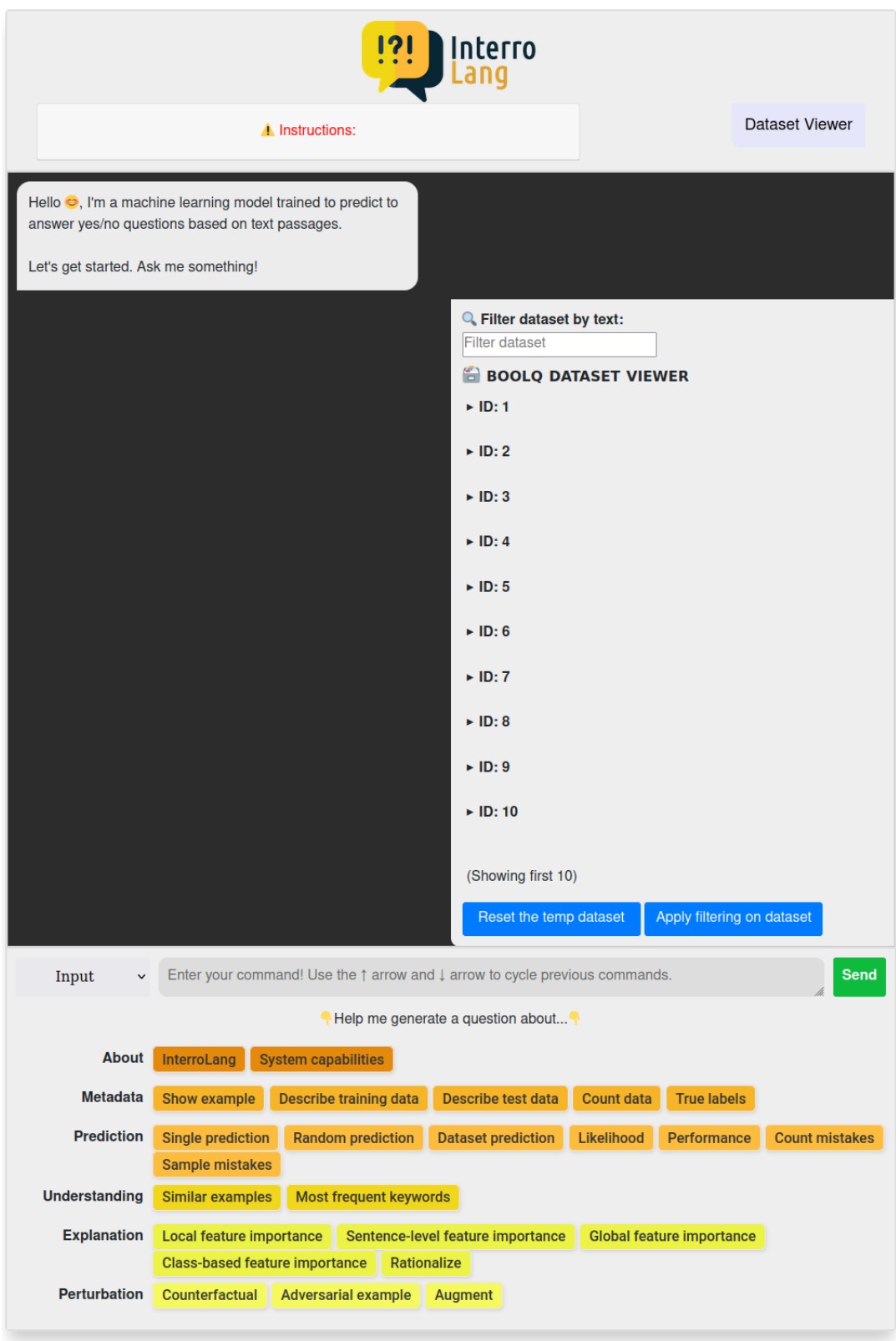

Figure 4: INTERROLANG interface with initial welcome message, opened dataset viewer (BoolQ) and sample generator buttons.

## Task 1

One after another, please try out each use case. For every dataset, we ask you to assign ratings for Correctness, Helpfulness and Satisfaction of **each** response from our tool. By clicking on "Feedback" of one response, you can rate each of them (👍/👎) on correctness, helpfulness and satisfaction. Please provide either 👍 or 👎 for each of these aspects.
Note: You can only provide feedback for the most recent message, so please rate each response before asking a follow-up question.

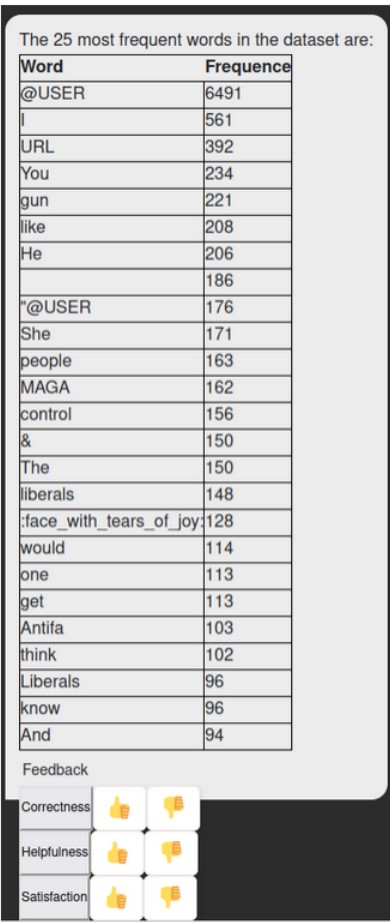

Figure 5: User study Task A1: Instruction.

## Task 2

After you have interacted with all three use cases, we ask you to provide **overall** ratings for the entire dialogues on Correctness, Helpfulness, Satisfaction and Fluency here:

Figure 6: User study Task A2: Instruction.

**BoolQ (Question Answering)**

Description (optional)

---

The dialogue with the question answering (BoolQ) system was **fluent**. *

|  | 1 | 2 | 3 | 4 | 5 |  |
|---|---|---|---|---|---|---|
| strongly disagree | ○ | ○ | ○ | ○ | ○ | strongly agree |

---

The dialogue with the question answering (BoolQ) system was **helpful**. *

|  | 1 | 2 | 3 | 4 | 5 |  |
|---|---|---|---|---|---|---|
| strongly disagree | ○ | ○ | ○ | ○ | ○ | strongly agree |

---

The dialogue with the question answering (BoolQ) system was **satisfying**. *

|  | 1 | 2 | 3 | 4 | 5 |  |
|---|---|---|---|---|---|---|
| strongly disagree | ○ | ○ | ○ | ○ | ○ | strongly agree |

---

The responses from the question answering (BoolQ) system seemed **correct**. *

|  | 1 | 2 | 3 | 4 | 5 |  |
|---|---|---|---|---|---|---|
| strongly disagree | ○ | ○ | ○ | ○ | ○ | strongly agree |

Figure 7: User study Task A2: Questionnaire on BoolQ.

For every dataset (BoolQ, DailyDialog, OLID), everyone gets **3 random IDs** assigned. [1]

> [1] There should be no duplicate IDs (except for between datasets), such that you can't see each others' guesses for any ID in the spreadsheet.

For each of the IDs, you **shuffle the list of operations** (see below), and this is considered as your itinerary.

**Explanation operations**

You can copy the following list and enter it in https://www.random.org/lists/ to randomize it:

Local feature importance (0)

Sent. feature importance (1)

Free-text rationale (2)

Counterfactual (3)

Adversarial example (4)

Similar examples (5)

You start with the first operation, **enter** it [2] and decide if this is sufficient for you to simulate the label.

> [2] You enter the corresponding question in the interface. If the wrong operation is parsed, you should not take the response of that wrong parse into account and try again, until you have the right one!

If it's not, you can **continue with the next operation** in the (randomized) list (the order has to be preserved for that ID!). It is possible to exploit a varying number of operations. That is, if you don't feel comfortable making a decision for the simulation, you can go through the entire list of six operations.

Based on the responses for that ID, you **choose the simulation label** and enter it in the spreadsheet.

Figure 8: User study Task B.

## H  Sample Dialogue (BoolQ with Adapter)

Hello 😊, I'm a machine learning model trained to predict to answer yes/no questions based on text passages.

Let's get started. Ask me something!

User: Hi! Which kind of a model do you use?

**General Information:**

| Name | Content |
| --- | --- |
| model_name | Distilbert-base-uncased-qa-boolq |
| model_summary | This model is a distilled version of the BERT base model. It was introduced in this paper. The code for the distillation process can be found here. This model is uncased: it does not make a difference between english and English. |
| task | Question Answering |

User: And what is the dataset?

**Training Data Details:**

| Name | Content |
| --- | --- |
| train_data_name | BoolQ Dataset: a question answering dataset for yes/no questions |
| train_data_source | plain text |
| train_data_language | English |
| train_data_number | 9427 |

The data contains information related to **Boolean question answering (yes/no)**.
The exact **feature names** in the data are listed as follows:

- question
- passage

The dataset has following **labels**:

- True
- False

The model scores *72.11% accuracy* on the data.

User: Can you show me how often the model makes incorrect predictions?

For **all** the instances in the data, the model is incorrect 912 out of 3270 times (error rate 0.279).
Here is the confusion matrix:

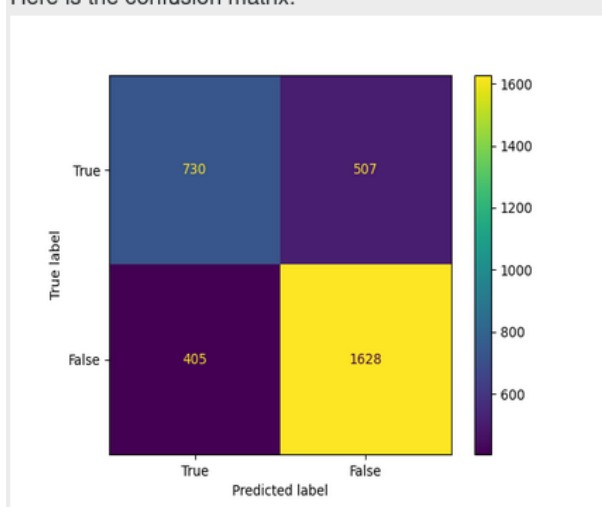

**User:** Can you show me some examples of the mistakes?

For **all** the instances in the data, the model is incorrect 912 out of 3270 times (error rate 0.279).

▼ Here are the ids of instances the model predicts incorrectly:

0, 10, 13, 19, 24, 33, 36, 38, 42, 45, 52, 55, 59, 60, 61, 62, 65, 66, 68, 71, 77, 79, 82, 84, 90, 91, 92, 93, 97, 101, 103, 108, 112, 113, 119, 122, 123, 127, 129, 130, 134, 140, 146, 154, 155, 159, 161, 164, 165, 166, 169, 172, 174, 182, 184, 188, 191, 198, 202, 213, 226, 228, 229, 233, 237, 259, 264, 268, 270, 271, 272, 275, 276, 284, 288, 294, 297, 301, 305, 321, 324, 326, 334, 339, 342, 343, 350, 360, 370, 387, 388,

**User:** Ok, great! What about the id 42? Can you show me this sample?

For the data with **id equal to 42**, the features are

question: is there an interstate that goes coast to coast
passage: U.S. Route 20 -- U.S. Highway 20 (US 20) is an east--west United States highway that stretches from the Pacific Northwest all the way to New England. The ``0'' in its route number indicates that US 20 is a coast-to-coast route. Spanning 3,365 miles (5,415 km), it is the longest road in the United States, and particularly from Idaho to Massachusetts, the route roughly parallels that of Interstate 90 (I-90), which is in turn the longest Interstate Highway in the U.S. There is a discontinuity in the official designation of US 20 through Yellowstone National Park, with unnumbered roads used to traverse the park.

**User:** Please show me the token attributions for this id.

**Question:** is there an interstate that goes coast to coast
**Passage:** U.S. Route 20 -- U.S. Highway 20 (US 20) is an east--west United States highway that stretches from the Pacific Northwest all the way to New England. The ``0'' in its route number indicates that US 20 is a coast-to-coast route. Spanning 3,365 miles (5,415 km), it is the longest road in the United States, and particularly from Idaho to Massachusetts, the route roughly parallels that of Interstate 90 (I-90), which is in turn the longest Interstate Highway in the U.S. There is a discontinuity in the official designation of US 20 through Yellowstone National Park, with unnumbered roads used to traverse the park.

Top 5 token(s): **disco ##nu that ##nti -**

▼ The visualization:

User: Can you do an adversarial attack?

**Label True (98.965%) --> False (99.675%)**

is there an interstate that **goes** coast to coast u . s . route 20 - - u . s . highway 20 ( us 20 ) is an east - - west united states highway that stretches from the pacific northwest all the way to

is there an interstate that **die** coast to coast u . s . route 20 - - u . s . highway 20 ( us 20 ) is an east - - west united states highway that stretches from the pacific northwest all the way to

User: Ok, thanks! Looks good :)

I'm glad I could help.

User: What would be the counterfactual for this instance?

- **[The original text]:** is there an interstate that goes coast to coast U.S. Route 20 -- U.S. Highway 20 (US 20) is an east--west United States highway that stretches from the Pacific Northwest all the way to New England. The ``0'' in its route number indicates that US 20 is a coast-to-coast route. Spanning 3,365 miles (5,415 km), it is the longest road in the United States, and particularly from Idaho to Massachusetts, the route roughly parallels that of Interstate 90 (I-90), which is in turn the longest Interstate Highway in the U.S. There is a discontinuity in the official designation of US 20 through Yellowstone National Park, with unnumbered roads used to traverse the park.
- **[Counterfactual 1]:** is there an interstate that **connect** coast **by** coast U. S. Route **25** - - U. S. Highway 20 ( US **2000** ) is an **end** - - west United States highway that stretches **throughout** the Pacific Northwest all the way to New England. The `` 0'' in its **plate** number indicates that US 20 is a coast - to - coast route. **On** 3, 365 miles ( 5, 415 km ), it is the longest road in the United States, and particularly from **Texas** to Massachusetts, the route roughly parallels that **numbering** Interstate 90 ( I - 90 ), which is in **fact** the longest Interstate Highway in the U. S. There is **this gap** in the **unofficial** designation of US 20 **at** Yellowstone National Park, with unnumbered roads used to traverse the park.

The predicted label **False** changes to **True**.

User: Can you show me the most important features overall (across all data )?

hypothetical, metaphor, avoided, problematic, unsuitable, shepard, limitation, dubious, ##utable, and scout are the most attributed.

User: Ok, I think that's it for today. Bye!

See you next time!