# OpenReview forum: "InterroLang: Exploring NLP Models and Datasets through Dialogue-based Explanations"
_EMNLP/2023/Conference — EMNLP 2023 Findings_

### Official Review · Reviewer_VaJL · 2023-07-21

**Soundness:** 3

**Excitement:**

3: Ambivalent: It has merits (e.g., it reports state-of-the-art results, the idea is nice), but there are key weaknesses (e.g., it describes incremental work), and it can significantly benefit from another round of revision. However, I won't object to accepting it if my co-reviewers champion it.

**Paper Topic And Main Contributions:**

This paper describes InterroLang, an extension to NLP models of the TalkToModel method for ML model explainability via a text-based dialogue interface. The authors implement InterroLang for three NLP tasks—dialogue act classification, question answering, and hate speech detection—with specific types of questions that can be asked, such as important features, counterfactual/adversarial examples, similar examples, and a free-form explanation. In a user study, the authors evaluate human perception and simulatability, demonstrating overall positive results.

The open-source software for InterroLang and its accompanying dataset are the primary contributions. InterroLang is a novel technique in the field of explainable AI for NLP, with its conversational interface and ability to generate rationales on multiple queries sequentially. The literature review and appendix provide thorough comparison to existing techniques. The paper itself is clear and well-written. The limitations are acknowledged, and sufficient detail is provided for the model and selected hyperparameters.

**Questions For The Authors:**

Question A: What is the computational cost for running these experiments?
Question B: Footnote 1 states that “while it might be tempting to use ChatGPT,...most interpretability methods require access to gradients, parameters or training data to make faithful explanations of their behavior…” However, in section 4, the implementation of Rationalization is ChatGPT and a few-shot GPT-like model. Aren’t these contradictory?
Question C: Section 3.2 states that a smaller model for QA is chosen because it is “more error-prone which increases the need for explanations.” Does this mean InterroLang is less useful for more powerful models, but wouldn’t you want it to still be useful for better models?
Question D: What is the intended use case for InterroLang?
Question E: Section 5.3.1 states that evaluation is performed on attribution in QA—what about the other types of explanations and tasks?

**Reasons To Accept:**

The open-source software for InterroLang and its accompanying dataset are the primary contributions. InterroLang is a novel technique in the field of explainable AI for NLP, with its conversational interface and ability to generate rationales on multiple queries sequentially. The literature review and appendix provide thorough comparison to existing techniques. The paper itself is clear and well-written. The limitations are acknowledged, and sufficient detail is provided for the model and selected hyperparameters.

**Reasons To Reject:**

The primary weakness is the user study. Only 10 students participated in the first part of the study, and four of the authors participated in the second half of the study. This is a small sample size; it is unclear whether IRB was in place, how students were recruited, or how much they were compensated; and the authors’ participation in evaluation seems like a potential source of bias. Finally, in a discussion of the results, it is stated that “one-off explanations alone are usually not sufficient or beneficial.” Although this intuitively makes sense, it is unclear exactly how this conclusion is drawn from the tables of results.

**Reproducibility:**

5: Could easily reproduce the results.

**Reviewer Confidence:**

4: Quite sure. I tried to check the important points carefully. It's unlikely, though conceivable, that I missed something that should affect my ratings.

**Typos Grammar Style And Presentation Improvements:**

•	I would add a few words about how the adapter method works.
•	I would mention the pre-defined questions from lines 128-129 in Fig. 1’s caption.
•	The title of the last row in Table 1 (NLU) doesn’t match the Similarity title in section 2.1.
•	Define Ren et al.’s PWWS
•	Clarify how NLTK’s stopwords are used to get the frequency set of keywords
•	Clarify what “perform the simulation” on line 399 means (e.g., guessing model output?)
•	“less parameters” -> “fewer parameters”
•	Give more explanation for why human satisfaction was the lowest category

---

> ### Author Rebuttal · Authors · 2023-08-28
>
> Thanks a lot for your comprehensive and thoughtful feedback! We appreciate your comments regarding the novelty of our InterroLang tool and study and the value of our open-source implementation as well as the dataset resulting from our user study. Thank you for also pointing out the extensive literature review and documentation in our appendix and that, overall, our paper is clearly written and provides sufficient detail for reproducibility.
>
> ### About Reason to Reject #1:
> We agree that conducting a large-scale user study for different explainability methods would be a very valuable contribution for the research community. However, due to the initially limited budget, it was out of scope of this submission. We are in the first stages of planning such a follow-up study, which requires a much more carefully designed setup involving frictionless execution on externally hosted GPUs, which we think is more fitting for a demo/industry track contribution.
> In our study, we tried to mitigate the bias as much as possible, e.g., by randomizing the sample IDs and using a randomized order of explanations for the simulatability study. A more in-depth analysis with more participants with different levels of expertise would require a separate study (see above). We will add the rank correlation as calculated by Kendall’s W (coefficient of concordance), i.e., how strongly the users agree on the ranking of operations given by Table 3 (subjective ratings).
> The main goal of InterroLang is to provide a tool that combines a variety of different approaches to explainability which can be used with different NLP models by the research community. We then provide an evaluation framework on top of it involving both objective and subjective ratings. We consider our study as a validation of InterroLang rather than a general evaluation of the explainability methods and their helpfulness. We believe that InterroLang (which we will continuously maintain as an open-source GitHub project) and the findings from our study will be useful for the research community and that it will enable larger studies and more in-depth evaluation of different explainability methods in a dialogical context.
> The participants of our user studies were recruited in-house: All of them were already working as research assistants in our institute and are compensated monthly based on national regulations. None of them had any prior experience with the explained models, but they were knowledgeable in NLP (l. 373).
>
> ### About Reason to Reject #2:
> Our conclusion that “one-off explanations alone are usually not sufficient or beneficial” is based on the results of the simulatability study summarized in Table 5. We found that an average number of turns needed to guess the model’s prediction varies based on which explanation types are shown in what order, but the number of turns is always much larger than one (between 3.70 and 4.14). Hence, we conclude that the user needs a longer conversation with the system and the evidence of multiple explainability methods to decide about model’s behavior.
>
> ### Question A:
> We have two different computational costs: One for preparing the parsing models and pre-computing explanations (requires a GPU); the other for running the tool with the pre-trained models. The latter can even run on customer-level CPUs. The execution time in this case is less than one minute for the biggest model (GPT-Neo), but only a few seconds when run on a GPU.
> For the best-performing parsing approach, we employed Adapters with BERT which requires less parameters and takes less memory than the other parsing models.
>
> ### Question B:
> We use ChatGPT only for (1) producing high-quality rationales to use in demonstrations (l. 301) and for (2) augmenting our intent recognition training data containing utterance-parse pairs (§2.2). We argue that these are legitimate use cases of ChatGPT.
>
> For almost every other part of InterroLang, ChatGPT is not applicable, though. This is mainly because feature attribution is not computable on a black-box model that is proprietary and hidden behind an API. In fact, all interpretability methods that rely on white-box access (gradients, parameters, training data) cannot be used in conjunction with ChatGPT. In preliminary tests using ChatGPT for generating the final response shown to users in the interface, we noticed factual errors (hallucinations/confabulations) inserted into the constituent parts used as input to ChatGPT. We find that there is no guaranteed correctness when NLG models are involved and in such dialogue settings, it might be better to follow rule-based responses to ensure factual correctness.
> InterroLang is a modular system and one of our goals is to have all modules be sourced from readily available tools. ChatGPT can easily be swapped with a sufficiently strong rationalizer and data augmenter, as soon as they become available open source. At the time of implementing InterroLang, however, we found that there is a large qualitative gap between ChatGPT and open-source LLMs (Dolly, GPT-Neo) and that’s why we opted to include it in these two parts of our framework.
>
> ### Question C:
> InterroLang does not require a small model or need the model to be “bad”. LLMs should work just as well, after we have implemented the necessary adaptations for sequence generation models (see Footnote 12 in the paper). We preferred more error-prone models, e.g., DistilBERT for QA, to examine some error cases and their associated explanations in InterroLang. We believe that we used the phrase “a smaller model” in a misleading manner. We will paraphrase it to clarify.
>
> ### Question D:
> In recent years, the field of interpretability has attracted researchers from many fields and continuously expanded the set of methods such as counterfactuals, rationales, local explanations and more. InterroLang provides an all-in-one system targeted at users beyond just interpretability experts and offers a simplified way (with a natural language interface instead of requiring users to code and fully comprehend the underlying methods) to examine the different types and model behaviors in one place.
>
> ### Question E:
> In §5.3.1, we took attribution and QA just as examples to illustrate the relation between use cases, explainability methods and the dialogue setting. In fact, the evaluation was done for all three tasks and all combinations between them and the explainability operations (see Tables 3 and 4 for more detail).
>
> We will further incorporate your valuable suggestions regarding presentation improvements in the camera-ready submission.

---

### Official Review · Reviewer_wK4Z · 2023-07-23

**Typos Grammar Style And Presentation Improvements:** N/A
**Soundness:** 4

**Excitement:**

4: Strong: This paper deepens the understanding of some phenomenon or lowers the barriers to an existing research direction.

**Missing References:**

N/A

**Paper Topic And Main Contributions:**

This paper adapt the converstaional explanation framework to the NLP domain and add new NLP-specific operations such as fre--text rationalization and illustrate its generalizability on three NLP tasks. Besides, this paper provided a reliable and useful tool to help fomulate these tasks.

**Questions For The Authors:**

1. I am not very clear about your proposed Helpfulness Rato, could you please illustrate for more details on this part?

**Reasons To Accept:**

1. This paper is very interesting, which provided a dialogue-based explanations to explore NLP models and datasets.
2. This work is solid, with a detailed and comprehensive tool, and detailed analysis and visulization results.
3. The paper is well-written, easy and clear to read and follow.
4. It provided a reliable and intersting tool, which has the potential to be widely used, especially incorporated with the existing LLMs.

**Reasons To Reject:**

1. Although this paper provided comprehensive and dense results, more details in the appendix could be moved to the experimental part.
2. In human evaluation, a consistency evalution between annotators should be considered, e.g., Kappa score.

**Reproducibility:**

4: Could mostly reproduce the results, but there may be some variation because of sample variance or minor variations in their interpretation of the protocol or method.

**Reviewer Confidence:**

3: Pretty sure, but there's a chance I missed something. Although I have a good feel for this area in general, I did not carefully check the paper's details, e.g., the math, experimental design, or novelty.

---

> ### Author Rebuttal · Authors · 2023-08-28
>
> Thank you for your review. You kindly pointed out the relevance of our study and of our tool as well as its level of detail regarding analyses and reporting of results. We are glad you also liked that our paper was easy and clear to read and follow and think that InterroLang has potential to be widely used in NLP research.
>
> ### About Reason to Reject #1:
> Based on your feedback, we will use the additional ninth page allotted to us for the camera-ready version to substantiate the main paper content with details from the appendix such as the Interface (App. F) and Dialogue Management (App. E).
>
> ### About Reason to Reject #2:
> Annotators were not guaranteed to ask about the same operations for the same instances in the same order, so the ratings cannot be directly compared to each other via Fleiss’ Kappa, which means we cannot compute inter-annotator agreement and instead we report average scores for each operation supported by InterroLang (Table 3). For the camera-ready version, we will add the rank correlation as calculated by Kendall’s W (coefficient of concordance), i.e., how strongly the users agree on the ranking of operations given by Table 3.
>
> ### Question:
> Our Helpfulness Ratio in our simulatability evaluation (Task B) refers to the number of “+1” helpfulness ratings divided by the sum of “+1” and “-1” helpfulness ratings. That is, for each operation and across all users, we first only consider those instances where the operation in question was used at all (non-zero helpfulness rating). Then we simply calculate the partition of instances which were assigned “+1” (helpful) ratings.
> As an example, let’s say our “Free-text rationale” operation has the following ratings from six different users (with one rating per user): [+1, 0, -1, +1, +1, +1].
> Helpfulness Ratio first removes all zeroes (unused), so we end up with [+1, -1, +1, +1, +1]. Then the result is 4/5 or 80% (all except one are positive).

---

### Official Review · Reviewer_4uXs · 2023-08-02

**Typos Grammar Style And Presentation Improvements:** 1. The explanation three dialogue act…
**Soundness:** 4

**Excitement:**

4: Strong: This paper deepens the understanding of some phenomenon or lowers the barriers to an existing research direction.

**Paper Topic And Main Contributions:**

The research paper proposed INTERROLANG, a user-centered, dialogue-based system for exploring three NLP datasets (DailyDialog, BoolQ and OLID) and model behavior. This system adapt the conversational explanation framework TALKTOMODEL to the NLP domain with new NLP-specific operations included, thereby allowing users to engage in interactive and dynamic conversations with the NLP model.  They conducted two user studies to evaluate the perceived correctness, helpfulness, and simulatability of the framework. A key finding from the user studies conducted reveals that one-off explanations are often not sufficient to provide users with a complete understanding of the model's output. Instead, users frequently require multiple exchanges or turns of conversation to accurately predict outcomes and comprehend the system's responses better. This contributes to the field of NLP explainability by emphasizing the importance of dialogue-based exploration over single-turn explanations.

**Questions For The Authors:**

Question A: Did you conduct any error analysis on the experiments results? What are the current challenges of your proposed INTERROLANG?

Question B: Could you provide more details about how human experiments are conducted? e.g. What are the exact question you asked the 10 students to evaluate correctness, helpfulness and satisfactions? How many samples are annotated and one sample is annotated by how many students?

Question C: In Table 3. Why does the "Show example" operation has such low CHS scores compared to the other operations?

**Reasons To Accept:**

1. This paper is very well written. The motivation, background, methodology, and model settings are clearly explained, which makes this paper has the potential to have more influence and reproducibility.
2. The user studies conducted in this research are comprehensive and detailed, offering both subjective (perceived correctness and helpfulness) and objective (simulatability) assessments of the dialogues.
3. The paper demonstrates the generalizability across three main NLP tasks (dialogue act classification, question answering, and hate speech detection) and open-sourced their tool for other models or datasets, making this research more approachable for further studies.

**Reasons To Reject:**

1. In the process of evaluating simulatability and helpfulness in Task B, it is worth noting that four out of six annotators were authors of the research themselves. This constitutes a significant majority of the evaluators which could introduce bias. The authors might implicitly favor their own model's behavior or interpret its outputs in a way that is aligned with their understanding and expectations (especially when evaluating helpfulness).

**Reproducibility:**

4: Could mostly reproduce the results, but there may be some variation because of sample variance or minor variations in their interpretation of the protocol or method.

**Reviewer Confidence:**

4: Quite sure. I tried to check the important points carefully. It's unlikely, though conceivable, that I missed something that should affect my ratings.

---

> ### Author Rebuttal · Authors · 2023-08-28
>
> We thank you for your review. We are glad that you recognized our paper’s contributions and found it to be very well written and explained. Thank you for mentioning our outcome to be both impactful and approachable for further studies due to its generalizability and for regarding our user studies to be comprehensive and detailed.
>
> ### About "Reason to Reject" #1:
> The introduction of bias into the simulatability evaluation (Task B) is a valid argument. We tried to mitigate the bias as much as possible and think it is reasonably small. This is because none of the annotators have an encompassing knowledge of the inner workings of the underlying model and, more importantly, have not previously looked at the IDs they were assigned. We randomized the IDs as well as the order in which explanations were to be selected. In this pioneering multi-turn simulatability study, we showed that our tool can be easily used for follow-up studies. A more in-depth analysis with more participants with different levels of expertise would require a separate study that we consider an important part of the future work.
>
> ### Question A:
> We performed an error analysis of the parsing accuracy (Table 2) for three different datasets (BoolQ, OLID, DailyDialog) and four parsing models/approaches that include Nearest Neighbors, GPT-Neo, FLAN-T5-base and BERT+Adapter. For the user study, we focused on evaluating correctness, helpfulness, satisfaction and fluency. Here, users could ask about any question for any instance from a given dataset, and since they were not guaranteed to ask about the same operations for the same instances in the same order, the ratings cannot be directly compared to each other. Hence, we cannot compute the inter-annotator agreement and instead we report average scores for each operation supported by InterroLang.
> Next to the plans mentioned in our future work paragraph (l. 538-554) (e.g., pro-actively guiding the user through the conversation and suggesting other relevant operations; using one LLM for all tasks; designing API endpoints; model refinement), our current challenges include implementing more interpretability methods (e.g., feature interactions, Chain-of-Thought rationales) and more filter types, while making the models more efficient and reliable on lengthy instances.
>
> ### Question B:
> In Task A, we ask the users to rate the responses in terms of correctness, helpfulness and satisfaction. Here are the annotation instructions for Task A:
>
> Task A1 (instance level):
> > “One after another, please try out each use case. For every dataset, we ask you to assign ratings for Correctness, Helpfulness and Satisfaction of each response from our tool. By clicking on “Feedback” of one response, you can rate each of them (👍/👎) on correctness, helpfulness and satisfaction. Please provide either 👍 or 👎 for each of these aspects.”
>
> Task A2 (dialogue level):
> > “After you have interacted with all three use cases, we ask you to provide overall ratings for the entire dialogues on Correctness, Helpfulness, Satisfaction and Fluency.” The rating scale headings then said: “The dialogue with the [question answering (BoolQ) | dialogue act classification (DailyDialog) | hate speech detection (OLID)] system [was fluent | was helpful | was satisfying | seemed correct].”
>
> For the camera-ready version, we will add the relevant content from the annotation instructions document we gave our annotators.
> We have only calculated the total number of dialogue turns annotated by users (1449 from Task A, 188 from Task B). We will report the number of distinct instances (= minus the overlap between users) in the final version.
>
> ### Question C:
> The "Show example" operation just displays one instance without providing any further explanations or insights into the model's behavior or why the instance is being classified in a specific way.

---

### Meta-Review · Area_Chair_rTbC · 2023-09-16

**Recommendation:** 2

**Metareview:**

This paper presents an approach for "interrogating" models by way of a conversational dialogue system.  A backend system supports several interpretability "operations", for things like running attribution methods, finding counterfactual explanations for an example, constructing adversarial inputs, rationalization, and more.  The implementation largely follows the ideas in TalkToModel, though the supported operations in the dialogue systems are different, and in some cases specialized to NLP.  It presents a fine-tuned model for interpreting user queries, essentially an intents-and-slots-style semantic parsing task to identify which interpretability operation to call. Performance of the parser is evaluated.

The paper then conducts end-to-end evaluation on helpfulness and simulatability.  Three datasets are used in its case study: DailyDialog for dialogue act classification, BoolQ (QA), and OLID (hate speech detection). Results are presented on parsing user intents, subjective ratings of system helpfulness, and simulatability, focused on evaluating different interpretation operations.

The reviewers found this paper interesting and well-written, with a comprehensive set of experiments. The idea of democratizing interpretability methods through techniques like this is very intriguing! However, there are a few significant issues with the work.

VaJL brings up another crucial question: "What is the intended use case for InterroLang?" The paper claims that the system is "user-centered". But there is very little discussion of why a dialogue system is needed or whether this is actually necessary or sufficient for the needs of its intended users. Given the underlying discrete set of operations that are supported, why not simply have a drop-down menu? I can see the argument that the dialogue system makes it more friendly to use, but does this actually have benefits? That is, are users without knowledge of the underlying operations able to use it "zero-shot" or do they need extensive documentation (e.g., a description of what operations are supported)? The trace in the appendix shows a fairly smooth interaction, but questions like "can you do an adversarial attack" or "what would be the counterfactual for this instance" do not seem like something a user would come up with on their own unless they knew what the model could do.

Related to this, another crucial issue is the lack of comparison to other systems.  The only real baselines are presented in Table 2 for the semantic parsing task (the nearest neighbors model, and weaker pre-trained models). Notably, the user study does not do A/B testing of the present system versus another possible system.  To justify the use of a dialogue system, I would expect to see a study where users are presented with an ablation of the system or some other kind of dataset visualizer to understand (a) are they able to do the tasks as effectively?  (b) Do they prefer the dialogue system to, say, a drop-down menu?

These two points (use case, baselines) were discussed privately and reviewers were left with lingering uncertainty about work, which is reflected in the revised ratings.

Furthermore, 4uXs points out that many of the user study participants are authors themselves. I believe this is also a weakness of the user study; although the tool may be intended for relative experts in machine learning, this is still not a representative sample.

---

### Decision · Program_Chairs · 2023-10-07

**Decision:**

Accept-Findings

**Comment:**

This paper presents an approach for "interrogating" models by way of a conversational dialogue system.  A backend system supports several interpretability "operations", for things like running attribution methods, finding counterfactual explanations for an example, constructing adversarial inputs, rationalization, and more.  The implementation largely follows the ideas in TalkToModel, though the supported operations in the dialogue systems are different, and in some cases specialized to NLP.  It presents a fine-tuned model for interpreting user queries, essentially an intents-and-slots-style semantic parsing task to identify which interpretability operation to call. Performance of the parser is evaluated.

The paper then conducts end-to-end evaluation on helpfulness and simulatability.  Three datasets are used in its case study: DailyDialog for dialogue act classification, BoolQ (QA), and OLID (hate speech detection). Results are presented on parsing user intents, subjective ratings of system helpfulness, and simulatability, focused on evaluating different interpretation operations.

The reviewers found this paper interesting and well-written, with a comprehensive set of experiments. The idea of democratizing interpretability methods through techniques like this is very intriguing! However, there are a few significant issues with the work.

VaJL brings up another crucial question: "What is the intended use case for InterroLang?" The paper claims that the system is "user-centered". But there is very little discussion of why a dialogue system is needed or whether this is actually necessary or sufficient for the needs of its intended users. Given the underlying discrete set of operations that are supported, why not simply have a drop-down menu? I can see the argument that the dialogue system makes it more friendly to use, but does this actually have benefits? That is, are users without knowledge of the underlying operations able to use it "zero-shot" or do they need extensive documentation (e.g., a description of what operations are supported)? The trace in the appendix shows a fairly smooth interaction, but questions like "can you do an adversarial attack" or "what would be the counterfactual for this instance" do not seem like something a user would come up with on their own unless they knew what the model could do.

Related to this, another crucial issue is the lack of comparison to other systems.  The only real baselines are presented in Table 2 for the semantic parsing task (the nearest neighbors model, and weaker pre-trained models). Notably, the user study does not do A/B testing of the present system versus another possible system.  To justify the use of a dialogue system, I would expect to see a study where users are presented with an ablation of the system or some other kind of dataset visualizer to understand (a) are they able to do the tasks as effectively?  (b) Do they prefer the dialogue system to, say, a drop-down menu?

These two points (use case, baselines) were discussed privately and reviewers were left with lingering uncertainty about work, which is reflected in the revised ratings.

Furthermore, 4uXs points out that many of the user study participants are authors themselves. I believe this is also a weakness of the user study; although the tool may be intended for relative experts in machine learning, this is still not a representative sample.